# Quantifying Social Interventions for Combating COVID-19 via a Symmetry-Based Model

**DOI:** 10.3390/ijerph20010476

**Published:** 2022-12-28

**Authors:** Lei Zhang, Guang-Hui She, Yu-Rong She, Rong Li, Zhen-Su She

**Affiliations:** 1Institute of Health Systems Engineering, College of Engineering, Peking University, Beijing 100871, China; 2State Key Laboratory for Turbulence & Complex Systems, Peking University, Beijing 100871, China

**Keywords:** symmetry analysis, order parameters, logistic dynamics, *SHR* dynamical model, COVID-19

## Abstract

The COVID-19 pandemic has revealed new features in terms of substantial changes in rates of infection, cure, and death as a result of social interventions, which significantly challenges traditional SEIR-type models. In this paper we developed a symmetry-based model for quantifying social interventions for combating COVID-19. We found that three key order parameters, separating degree (S) for susceptible populations, healing degree (H) for mild cases, and rescuing degree (R) for severe cases, all display logistic dynamics, establishing a novel dynamic model named *SHR*. Furthermore, we discovered two evolutionary patterns of healing degree with a universal power law in 23 areas in the first wave. Remarkably, the model yielded a quantitative evaluation of the dynamic back-to-zero policy in the third wave in Beijing using 12 datasets of different sizes. In conclusion, the *SHR* model constitutes a rational basis by which we can understand this complex epidemic and policymakers can carry out sustainable anti-epidemic measures to minimize its impact.

## 1. Introduction

In late 2019, the novel coronavirus broke out and caused over 545 million infections worldwide, with more than 6.3 million deaths as of 1 July 2022. COVID-19 has attracted a great deal of attention, and the World Health Organization (WHO) regularly publishes detailed information on changes in the global epidemic, providing data for deep modeling of infection dynamics to reveal regularities in the epidemic’s evolution. A good mathematical model should be able to explain great variation across countries and predict future trends of the epidemic to provide information for policymakers to implement measures to combat it [1,2].

Many models have been developed to simulate the spread of epidemics with similar mathematical structures, dating back as far as the 18th century French mathematician Daniel Bernoulli (1700–1782) [3]. The Scottish scientists Kermack and McKendrick proposed the famous compartment model in 1927 [4], which divides people into three categories: susceptible, infectious, and recovered. For infectious diseases such as coronavirus pneumonia, which have a certain incubation period, a new category, exposed, is specifically designated for people who have been exposed but are not yet infected. Assuming that the population is evenly mixed, individuals in the same category are transferred to the next category at the same rate of transition. This is currently the most widely used SEIR model. Early in COVID-19, many research teams applied the SEIR model to predict the evolution of the epidemic [5,6].

However, social interventions have played an important role during the evolution of this epidemic [7], and different nations have shown different evolutionary patterns because they have adopted different policies. From the published results, it is clear that the traditional SEIR model is unable to model national responses to COVID-19 [8].

Three main improvements have been made to the SEIR model. First, new categories were added to simulate a more realistic population distribution [9,10,11,12,13,14,15,16,17]. For example, López et al. [9] added “confined” populations to simulate the effects of different unblocking policies. However, the reliability of the model is unclear when new parameters are introduced [8,18]. Second, the originally assumed constant transition rates were adjusted to a time-dependent function to fit the real data [7,19,20,21,22,23,24,25,26,27]. For example, Cheynet et al. [20,21,22,23,24], expanding on the work of Peng et al. [7], used multiple functional forms to fit cure and mortality rates. These improvements are somewhat arbitrary and marginal in their ability to reveal universal patterns in the evolution of the epidemic, although they can describe a portion of the epidemic data. Third, the agent-based model [28], and the consideration of meteorological changes [8], transport networks [29], and population mobility data [30] yield a more realistic mix of people for modeling, but these models usually require large amounts of data to determine the parameters, and then it is difficult to make precise predictions with them [1]. Thus, there is an urgent need to develop a novel dynamical system model that is consistent with the actual evolution of the epidemic and can reveal the effects of social interventions.

In this paper, we developed a symmetry-based model of COVID-19 that considers social interventions as a new class of symmetry-breaking in addition to the usual symmetry-breaking processes already modeled by the SEIR. This novel class of symmetry-breaking is described by three order parameters with logistic dynamic behavior: the separating degree (S) for susceptible populations, the healing degree (H) for mild cases, and the rescuing degree (R) for severe cases. The model was applied to explain actual epidemic data from Italy and 24 areas in China in the first wave, which display quantitatively different patterns under different intervention policies. Furthermore, we discovered two evolutionary patterns of healing degree with a universal power law. A remarkable outcome of the model is a quantitative evaluation of the so-called dynamic back-to-zero policy in the current third wave in Beijing.

The main contribution of this paper is to determine that the three transformation rates have a dynamic logistic behavior, which makes it possible to compare and summarize the universal laws of the epidemic across countries. In the future, by establishing complex relationships between these three groups and other populations, such as the interval observers and the immunized group, more realistic epidemic evolution can be simulated. The model constitutes a rational basis by which we can understand this complex epidemic and policymakers can carry out sustainable anti-epidemic measures to minimize its impact on people and society.

## 2. Mathematical Model

First, we considered the outbreak process to be composed of a series of symmetry-breaking events and then used symmetry analysis from statistical physics to develop a mathematical model to portray the multiple symmetry-breaking processes during the spread of COVID-19. The approach, which originated from the famous physicist Landau, is to define relevant order parameters with simple underlying mathematical structure due to the self-organization principle [31]. We previously applied this approach to the study of COVID-19 mortality [32], wall turbulence [33,34], and high-temperature superconductivity [35], with very fruitful outcomes.

### 2.1. Symmetry Analysis

During the spread of COVID-19, there were two opposing processes: the spread of the new virus among the population and the social interventions to limit the spread of the virus and allow people to heal. These two processes constitute multiple symmetrical breakings in the population dynamics (see Figure 1).

#### 2.1.1. Virus Transmission

First, there was symmetry breaking caused by the spread of a new virus (see Figure 1a). In the absence of the new virus, the population system is uniform, i.e., there is symmetry. When the new virus (SARS-CoV-2) appears, a population called “infected” emerges, i.e., symmetry-breaking occurs, with non-uniformity appearing in the population. The model that divides the population into susceptible, exposed, recovered, etc., is a way to describe this non-uniformity, which is quantified by several transition rates. For example, a susceptible person will become an exposed person at transition rate a. When a=0, no transitions between two groups occur, and the epidemic will not spread. When a=100%, all susceptible individuals will be infected when they come into contact with an infected person. Similarly, exposed individuals will become infected at transition rate b, who in turn will recover at transition rate c. Finally, the equation can be written based on the conservation of total populations. This is the famous SEIR model viewed from a symmetry-breaking perspective.

#### 2.1.2. Social Interventions

When large-scale social interventions are implemented, the transition process between groups is affected and new symmetry breaking from “free transmission” to “restricted transmission” occurs, as shown in Figure 1b. Social interventions can be divided into three main kinds: separating measures for susceptible populations, healing measures for mild cases, and rescuing measures for severe cases.

The first type of intervention, separating measures, targets susceptible people, and includes city lockdown, social distancing, wearing masks, washing hands, etc., to block the free transmission of the virus. We therefore define the separating degree (S) to quantify the intensity of the separating measures. When S=0, no separating measure is imposed and the virus is free to spread. When S=100%, the strictest separating measures are adopted, and the virus cannot spread at all.

Similarly, we define the healing degree (H) to quantify the intensity of healing measures, including increasing the number of medical staff and beds (e.g., Fang-Cang hospitals), promoting standard treatment protocols, etc., which favors the transition of patients back to the healthy population. Finally, the rescuing degree (R) is defined to quantify the intensity of rescuing measures, including the addition of ICU beds and life-support equipment, which allows a portion of severe patients to be transferred back to the healthy population.

### 2.2. SHR Model

The three order parameters have simple dynamics, as described here. In the early stages of the epidemic, the new virus is in a state of free spread, and the separating degree (S), healing degree (H), and rescuing degree (R) are zero. As the outbreak progresses, these three types of interventions begin to grow exponentially, as society responds to the epidemic. However, due to finite social resources, the growth rate of these interventions will slow down and eventually reach a saturation point. Therefore, the three order parameters satisfy the following logistic dynamics equations:(1)$$\frac{dS}{dt}=\gamma \left(1-\frac{S}{S_{1}}\right)S$$
(2)$$\frac{dH}{dt}=\alpha \left(1-\frac{H}{H_{1}}\right)H$$
(3)$$\frac{dR}{dt}=\mu \left(1-\frac{R}{R_{1}}\right)R$$
where $S_{1}$ is the saturation separating degree $\left(0\leq S\leq S_{1}\right)$, $H_{1}$ is the saturation healing degree $\left(0\leq H\leq H_{1}\right)$, and $R_{1}$ is the saturation rescuing degree $\left(0\leq R\leq R_{1}\right)$; γ is the initial growth rate of separating degree S, α is the initial growth rate of healing degree H, and μ is the initial growth rate of rescuing degree R.

Equations (1)–(3) are often used to describe the evolution of biological systems at different scales, such as the growth of microorganisms or biological populations, the spread of an epidemic, the growth of human weight and height with age, and changes in the human photon radiation signal with age [32,36,37,38]. In this paper, we demonstrate that logistic growth can also quantify social interventions, as validated below by our accurate predictions of various populations from WHO-reported epidemic data.

Now, let us connect the three order parameters to familiar quantities in epidemic modeling, namely infection rate ($V_{I}$), cure rate ($V_{C}$), and death rate ($V_{D}$), which are defined as
(4)$$V_{I}(t)=\frac{1}{Q(t)}\frac{dI(t)}{dt}$$
(5)$$V_{C}(t)=\frac{1}{Q(t)}\frac{dC(t)}{dt}$$
(6)$$V_{D}(t)=\frac{1}{Q(t)}\frac{dD(t)}{dt}$$
where Q(t) is the quarantined cases still in hospitals, I(t) is the reported infected cases, C(t) is the reported cured cases, and D(t) is the reported deaths, which satisfy the conservation of numbers:(7)$$\frac{dQ(t)}{dt}=\frac{dI(t)}{dt}-\frac{dC(t)}{dt}-\frac{dD(t)}{dt}$$

Note that in the early stages of the epidemic, when the intensity of social interventions is zero, the virus is free to spread, and the infection rate ($V_{I}$) and death rate ($V_{D}$) are constant, named $V_{I}^{0}$ and $V_{D}^{0}$, respectively. The cure rate ($V_{C}$) is initially zero naturally, because a finite period is needed for patients to recover. As social interventions intensify, the infection rate ($V_{I}$) and death rate ($V_{D}$) eventually decay to a plateau, while the cure rate ($V_{C}$) grows to a saturation value. Therefore, we have the following simple relations:(8)$$V_{I}(t)=V_{I}^{0}\left(1-S(t)\right)$$
(9)$$V_{C}(t)=V_{C}^{0}\times H(t)$$
(10)$$V_{D}(t)=V_{D}^{0}\left(1-R(t)\right)$$

Thus, Equations (1)–(10) make up the new infectious disease dynamics model called *SHR*. In contrast to the traditional SEIR model, the *SHR* model focuses not only on changes in various population groups but also on the impact of social interventions on the epidemic.

As shown in Figure 2, changes in social interventions will lead to changes in the intensity of separating degree (S), healing degree (H), and rescuing degree (R), which will be further reflected in the corresponding infection rate ($V_{I}$), cure rate ($V_{C}$), and death rate ($V_{D}$). Ultimately, what comes to light is the daily variation in reported data for infected cases (I(t)), cured cases (C(t)), and deaths (D(t)). We then have a picture of the evolution of the epidemic: three populations are determined by two pairs of dual degrees. On the one hand, virus action causes infection and healing degree brings recovery; on the other hand, virus action causes death and rescuing degree brings life back.

### 2.3. Analytic Solution

The three order parameters (Equations (1)–(3)) have logistic functions as their analytical solutions, as follows:(11)$$S(t)=\frac{S_{1}}{1+e^{-\gamma \left(t-t_{s}\right)}}$$
(12)$$H(t)=\frac{H_{1}}{1+e^{-\alpha \left(t-t_{h}\right)}}$$
(13)$$R(t)=\frac{R_{1}}{1+e^{-\mu \left(t-t_{r}\right)}}$$
where $t_{s}$, $t_{h}$, and $t_{r}$ are respectively the characteristic times at the midpoints of the three logistic functions S, H, and R.

In the following, we will discuss the meaning of these model parameters. Taking separating degree S as an example:

When $t\gg t_{s}>0$, Equation (11) can be written as
(14)$$S\left(t\right)\approx S_{1}$$

Therefore, $S_{1}$ denotes the saturated separating degree in the late epidemic period.

When $t=t_{s}$, Equation (11) can be written as
(15)$$S\left(t_{s}\right)=\frac{S_{1}}{2}$$

Therefore, $t_{s}$ denotes the characteristic time when the separating degree reaches half of its saturation value.

When t≈0, $0<S\left(t\right)\ll S_{1}$, Equation (1) can be written as
(16)$$\gamma =\underset{t\to 0}{\lim}\frac{S{\left(t\right)}^{'}}{S\left(t\right)}$$

Therefore, γ represents the initial exponential growth rate of the separating degree at the early stages of the epidemic, or the initial social response speed to the virus spreading.

When t≈0, S(t)≈0, Equation (8) can be written as
(17)$$V_{I}(t)\approx V_{I}^{0}$$

Therefore, $V_{I}^{0}$ indicates the initial infection rate when the virus is freely spreading in the early stages of the epidemic.

The above explanation involving separating degree S can be reproduced for healing degree H and rescuing degree R.

Furthermore, we can derive a log law for approximately determining the healing degree. Specifically, if we take an early time t=1, the initial healing degree $H_{0}$ satisfies Equation (12):(18)$$H_{0}=\frac{H_{1}}{1+e^{-\alpha \left(1-t_{h}\right)}}$$

Most data indicate$\;H_{1}\gg H_{0}$ and $t_{h}\gg 1$, so the above equation can be simplified as
(19)$$\alpha \approx \frac{1}{t_{h}}ln\left(\frac{H_{1}}{H_{0}}\right)$$

Therefore, growth rate α of the healing degree is inversely proportional to its inflection point $t_{h}$, with a coefficient determined by $\ln(H_{1}/H_{0})$.

The system of the differential equation for the transmission of an epidemic given above can be transformed into the difference equation to predict daily reported data. This yields the following set of equations that characterize the spread of the epidemic:(20)$$\Delta I\left(i\right)=V_{I}^{0}\times \left(1-S(i)\right)\times Q\left(i-1\right)$$
(21)$$\Delta C\left(i\right)=V_{C}^{0}\times H\left(i\right)\times Q\left(i-1\right)$$
(22)$$\Delta D\left(i\right)=V_{D}^{0}\times \left(1-R(i)\right)\times Q\left(i-1\right)$$
(23)Q(i)−Q(i−1)=ΔI(i)−ΔC(i)−ΔD(i)
where i represents the day, and ΔI, ΔC, and ΔD represent the number of daily infected (confirmed), cured, and dead on that day, respectively. The three types of order parameters, separating degree (S), healing degree (H), and rescuing degree (R), are taken as the analytic solutions for the corresponding logistic functions, i.e., Equations (11)–(13).

### 2.4. Parameter Determination

We adopted a two-step parameter inversion process. The first step is to carry out least squares fitting of the actual rate data by Equations (4)–(6), which independently yield three sets of parameters: $\left\{V_{I}^{0},\;\gamma ,\;S_{1},\;t_{s}\right\}$, $\left\{\alpha ,\;H_{1},\;t_{h}\right\}$, and $\left\{V_{D}^{0},\;\mu ,\;R_{1},\;t_{r}\right\}$. Here, we set $V_{C}^{0}=1/\mathrm{day}$ to reduce the redundancy of the parameters. Since these three sets of parameters are independent of each other, the parameter uncertainty is very small (see Table 1).

The second step is to fit the case data by the prediction of Equations (20)–(23), with fine-tuning of the parameter values around those obtained in the first step. This consists of minimizing errors in the predictions of all case data (new and accumulated), then iterating and updating the parameters until the difference in the regression standard deviation between the two cycles is less than $10^{-6}$, to yield the best parameter set. This would eliminate the effect of singular events in the case data.

We applied the model to describe the epidemic data from Italy and 24 provinces (municipalities) in China in the first wave, whose total confirmed cases exceeded 100 (to be statistically meaningful), according to the publicly available epidemic database at Hopkins University [39] and the Hubei Province Health Commission [40]. The model parameters are shown in Table 1, Table 2, Table 3 and Table 4.

All data and code used in this study are publicly available at https://github.com/zhanglei-pku/COVID-19-SHR-model (accessed on 26 December 2022).

## 3. Results

### 3.1. Quantitative Comparison of Social Intervention Degrees

Hubei Province and Italy, as the centers of the initial outbreaks in Asia and Europe, respectively, in the first wave of the epidemic, have similar dynamics and are comparable in terms of population, area, and latitude. Therefore, we applied the *SHR* model to quantitatively compare the evolutionary characteristics of the epidemic in these two regions to reveal the dominant role of social interventions.

First, Figure 3 shows a complete representation of the evolution of the order parameters (Figure 3a–c), the transition rates (Figure 3d–f), and the population numbers (Figure 3g–i). Taking the separating process as an example, separating degree S leads to a decrease by 2–3 orders of magnitude in the corresponding infection rate (Figure 3d), which manifests in the number of daily confirmed cases, giving rise to growth–decay behavior (Figure 3g). Similarly, both the healing and rescuing processes obey this pattern. Close agreement between the data and predicted curves validates the model.

Second, we found that the initial values of all three transition rates were very close in the early stages of the epidemic (see Figure 3d–f), with an initial infection rate of about 0.5, a cure rate of about 0, and a death rate of about 0.03. Since the virus strains in Italy and Hubei were the same in the first wave, the close transition rates validate the *SHR* model’s implication that the early evolutionary behavior was dominated by the free spread of the virus (see the “Mathematical model” section). More importantly, these key constants reflect the ability of the virus in terms of infectivity, lethality, etc. It would be worthwhile in the future to use this model to conduct comparison studies of different mutant strains or viruses.

Finally, the *SHR* model demonstrates the dominant role of social intervention in affecting the direction of the epidemic. Taking the separating process as an example, as Figure 3a shows, the initial value of separating degree $S_{0}$ in Italy was higher than that in Hubei Province, which is related to the fact that Italy was alerted by the outbreak in Hubei. However, the growth rate of separating degree γ in Hubei is 1.8 times higher than that in Italy (see Table 1), suggesting the effect of cultural differences and the intensity of government interventions. For example, according to press reports, Italy’s lockdown policy was late, with the lockdown of the Lombardy region only starting when the number of quarantined cases reached 6387 (8 March), with separating degree S=44.1%, whereas Hubei Province closed Wuhan when the number of quarantined cases reached 494 with separating degree S=10.7%. The smaller this value is, the earlier the society acted against the disease. Thus, after 20 days, Hubei’s separating degree surpassed Italy’s, and eventually, the cumulative number of confirmed cases in Italy (233,000) was 3.4 times that in Hubei (68,000). This shows that the *SHR* model is very effective at quantifying the intensity of the separating degree of the epidemic.

More interestingly, the *SHR* model provides a quantitative framework for scientifically selecting the timing of unblocking and predicting the likelihood of a secondary outbreak. As shown in Figure 3a, separating degree S of Hubei Province was close to 100% after 65 days, with an almost zero infection rate, and Wuhan was successfully decontrolled after 79 days. In contrast, saturation separating degree $S_{1}$ in Italy eventually increased to 98%, which resulted in a small but non-zero infection rate of around 1%; this small rate, however, yielded a long infection process in Italy until the second outbreak. Therefore, saturation separating degree $S_{1}$ is an important parameter for determining when the epidemic ends.

Similarly, for both the healing and rescuing process, the *SHR* model allows us to quantitatively evaluate the role of the corresponding social interventions. For example, if the growth rate of Italy’s healing degree α (0.03) was the same as Hubei’s (0.08) and all other parameters remained the same, the model’s simulation results show that during the first wave in Italy, about 75,000 people would have been spared from infection and about 15,000 from death. If the growth rate of Italy’s rescuing degree μ (0.09) reached Hubei’s value (0.16), about 14,000 deaths would eventually be avoided. This shows the dominant role of social interventions in the evolution of the epidemic.

### 3.2. Two Evolutionary Patterns of Healing Degree in 23 Areas

Modeling the dynamics of wider outbreak spillover areas under well-controlled conditions, as compared to outbreak centers such as Hubei and Italy, can reveal different patterns in social interventions. The spillover areas generally had lower death rates because the healthcare systems were alerted. For example, there were only 4 deaths on average across the 30 provinces (municipalities) in mainland China, excluding Hubei, in the first wave of the epidemic. For these areas, the *SHR* model can be reduced by setting the initial mortality rate $V_{D}^{0}$ to zero so that the equations can be analytically solved. We compared the solution with the epidemic data from the 23 spillover areas with the total of confirmed populations exceeding 100 in the first wave (see Figure 4).

First, in contrast to the outbreak centers, the governments in the spillover areas responded more quickly, and the average growth rates of separating (γ) and healing (α) were both 1.5 times that of Hubei Province, which resulted in the inflection point occurring 11.3 and 6.6 days earlier than Hubei, respectively. As a result, these areas achieved impressive anti-epidemic performance with an average of 27 ± 5 days to clear and only 528 ± 395 infections (see Table 2).

Next, we found that 17 of the 23 areas with whole evolutionary cycles showed a specific feature: the growth rate of healing degree α had a power law dependence on its inflection point $t_{h}$ (see Figure 5a). This universal power law expresses that the whole country follows a uniform guideline for fighting the epidemic. Interestingly, the 17 areas follow two different power laws, because of two different coefficients $H_{1}/H_{0}$ (see Equation (19) ), as displayed in Figure 5b. The first evolutionary pattern, involving 8 areas, had a relatively greater ratio of late to early healing degree ($H_{1}/H_{0}$), approximately 3 times that of pattern 2 (N = 9). Thus, we believe that the ratio $H_{1}/H_{0}$ is an indicator of how effective local interventions are. Indeed, this is confirmed by the data of cured cases. As shown in Figure 5c, areas of pattern 1 have a shorter duration of 36 days (IQR, 34, 38), with the healing process ending on average one week earlier than in pattern 2. These results then demonstrate that the present model is effective at describing the impact of social interventions on the evolution of epidemics.

### 3.3. Beijing’s Successful Experiences in Two Waves

We also applied the *SHR* model to quantitatively characterize the evolution of different epidemic waves to reveal the subtle changes in government interventions. For comparison, we selected the first wave in Beijing in January 2020 and the second wave in June of the same year. The two outbreaks were comparable and had the same early virus strain. The difference is that the first wave exhibited a localized spillover pattern due to the migration from Hubei Province. The second wave was a central outbreak due to the spread of cases from the Xinfadi market, similar to the Wuhan Huanan Seafood Market. We used the *SHR* model to simulate both waves and obtained good data fit, as shown in Figure 6.

First, the initial values of separating degree $S_{0}$ and healing degree $H_{0}$ were greater for the first wave than the second wave (see Figure 6a,d). This is consistent with the fact that Beijing quickly launched a first-level response to the first wave of the outbreak, while the initial interventions were less intense during the second wave. However, the growth rate of separating degree γ in the second wave was 2.3 times that in the first wave, so by the ninth day, the separating degree exceeded that in the first wave and rapidly approached saturation $S_{1}$ of 100%. Similarly, the growth rate of healing degree α in the second wave was 8.5 times that in the first wave, thus approaching saturation $H_{0}$ at around 20 days (see Figure 6d). This suggests that Beijing intensified the interventions in the second wave in a shorter period than it did in the first wave.

Second, we note that the initial infection and cure rates in both waves were very close (see Figure 6b,e), consistent with the fact that the virus strain was the same. As the epidemic spread, we see that the infection rate in the second wave decreased more rapidly by almost three orders of magnitude as a result of social interventions, thus new confirmed cases were successfully cleared out 5 days earlier than in the first wave (see Figure 6c,f and Table 3).

In the end, the cumulative number of infections in the second wave was 335, less than the 395 in the first wave, with no deaths, which makes it a case of remarkably successful social interventions. If the second wave had the same parameters as the first wave, Beijing would have had 22,000 more infected people, which could have led to a much more serious impact on the health system. For this reason, the *SHR* model can quantitatively evaluate improvements in social interventions.

### 3.4. Simulation of the Impact of Dynamic Back-to-Zero Policy

At the end of 2021, in the face of the highly transmissible and insidious nature of the Omicron variant of the novel coronavirus, the Chinese government decided to adopt a dynamic back-to-zero policy. This policy showed impressive results in the control of outbreaks in Jilin and Shenzhen in 2022. However, for super-large cities with a population exceeding 20 million, such as Shanghai and Beijing, the policy would face a much bigger challenge. It would be interesting to quantitatively evaluate the impact of the dynamic back-to-zero policy, to help policymakers better balance epidemic prevention/control and economic activities/decontrol. Therefore, we applied the *SHR* model to the third wave of the outbreak in Beijing on 22 April 2022 to evaluate the evolution of the epidemic under different control intensities.

First, from actual reported data, the evolution of the third wave in Beijing has shown very different characteristics from the previous two waves. For example, the number of daily confirmed cases (excluding asymptomatic infections) in the third wave exceeded 50 in less than a week, surpassing the previous record high of 44. Furthermore, the peak was not followed by a rapid containment as in the previous two waves, but by a plateau period that lasted for more than 20 days, with more complex features, such as a double peak. The infection rate increased to about 6% and then entered a phase of fluctuation (see Figure 7b), which is one order of magnitude greater than the 0.4% in the first two waves (see Figure 6b), indicating the superinfection of Omicron. These characteristics pose a serious challenge to our ability to explain and predict the third epidemic wave by traditional models.

Yet, the *SHR* model gives a quite good simulation and a reasonable explanation of the evolution. First, the simulations using epidemic data (as of 23 May) show that the current separating degree varies around saturation $S_{1}$ at 96.4% for 23 May (see Figure 7a), compared to the previous two waves, when the separating degree rose rapidly to 100%. This may reflect a sense of fatigue after two waves, so the dynamic back-to-zero policy adopted a degree of flexibility. On 23 May, with intervention parameters determined on that day, the *SHR* model predicted that the number of daily confirmed cases would drop below 10 around 15 June, with a cumulative total of 1891 confirmed cases (see Figure 7c). If a moderate relaxation of control was considered with a smaller saturation separating degree of $S_{1}=93\%$, the time under control would be delayed by 3 weeks (6 July) at the cost of an additional 804 cumulative confirmed cases. However, if the intensity was further increased with saturated separating degree $S_{1}=98\%$, then the time under control might come 10 days earlier (6 June) and the cumulative confirmed cases would be reduced to 1572. Note that in Beijing, the actual saturated separating degree $S_{1}$ was 96.8% and the cumulative confirmed cases totaled 1655 after a series of dynamic adjustments. We can see that the *SHR* model can simulate the results of adopting different intensities of separating measures to a certain extent, and saturated separating degree $S_{1}$ is an effective parameter to quantify such intensity.

To check the robustness of the above predictions, we applied the *SHR* model to epidemic data with different ending dates between 12 and 23 May and formed 12 datasets of different sizes (see Figure 7d,e). As shown in Table 4, the model parameters performed very robustly over the 12 consecutive days of prediction; for example, the key saturated separating degree $S_{1}$ fluctuating only between 96.4 and 96.7% (see Figure 7d). It is worth pointing out that this model is an inverse derivation of saturated separating degree $S_{1}$ based on the epidemic data, and it would be interesting to investigate the conditions to directly estimate the model parameters from social control policies.

As shown in Figure 7e, the separating degree on 12 May was 96.5%, a relatively low value, and reached a maximum $S_{1}$ of 96.7% on 15 May, and the predicted cumulative number of confirmed cases was 1983. In the end, the actual evolution was very close to the predicted results as of 21 May, which very accurately portrays the evolutionary behavior of the cumulative cases for 20 consecutive days after 21 May, with a mean relative error of only 3% (see Figure 7e). However, it is only through the daily trajectory of separating intensity shown in the *SHR* model that it becomes clear that this is the result of Beijing’s flexible adjustment to the separating degree (see Figure 7d).

## 4. Conclusions

In the present work, we attempted to reveal the mathematical structure associated with social interventions that had never been so intensive before. Based on symmetry analysis, this paper adds two or three new processes generated by social interventions to the epidemic evolution beyond the traditional SEIR model. In particular, we introduce three key order parameters: separating degree (S) for susceptible populations, healing degree (H) for mild cases, and rescuing degree (R) for severe cases, thus establishing a new dynamical model, *SHR*. The epidemic patterns under different intervention policies are displayed quantitatively by applying the *SHR* model to actual epidemic data from Italy and 24 areas in China in the different waves. Furthermore, two evolutionary patterns of healing degree with a universal power law are discovered. To better balance epidemic prevention/control and economic activities/decontrol, the model was used to simulate the evolution of the so-called dynamic back-to-zero policy in the current third wave in Beijing. Based on an understanding of the complex epidemic provided by the model, policymakers can carry out sustainable anti-epidemic measures to minimize its impact on people and society.

## Figures and Tables

**Figure 1 ijerph-20-00476-f001:**
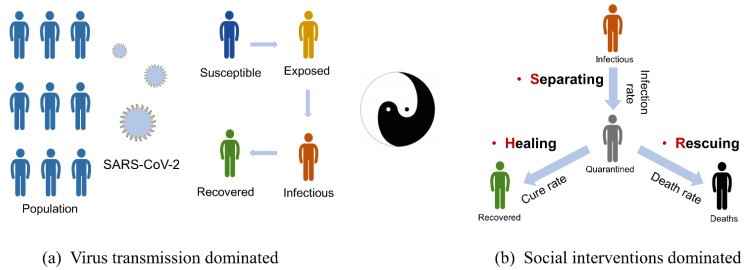
Two processes constituting multiple symmetrical breakings in population dynamics. Blue indicates susceptible population, light yellow indicates exposed population, orange is infected cases, green is recovered cases, gray is severe cases, and black is deaths. (**a**) Popular SEIR model viewed from a symmetry-breaking perspective. (**b**) Three types of social interventions led to symmetry breaking in corresponding populations.

**Figure 2 ijerph-20-00476-f002:**
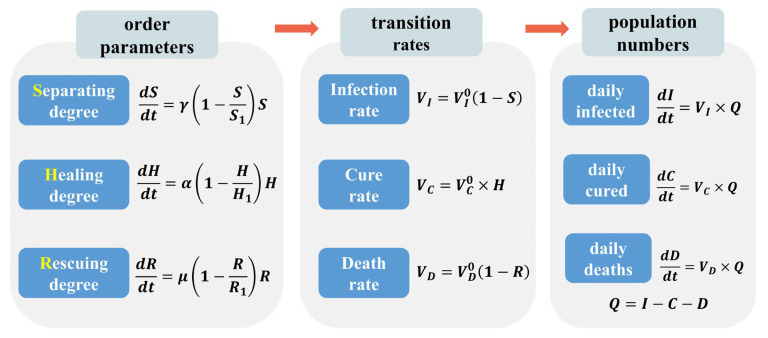
Schematic representation of *SHR* model.

**Figure 3 ijerph-20-00476-f003:**
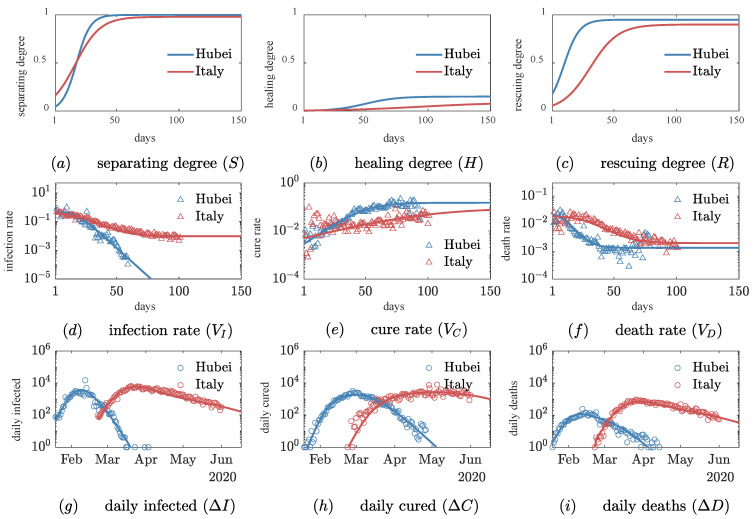
Comparison of evolution of first wave of the epidemic in Hubei Province and Italy. Red and blue circles indicate data collected by Hubei Provincial Health Commission and Johns Hopkins University real-time epidemic surveillance system, respectively (as of 2 June 2020), triangles indicate resulting calculated rate data, and solid line indicates simulation results of *SHR* model. Simulation parameters are shown in Table 1. (**a**) Separating degree, (**b**) healing degree, and (**c**) rescuing degree; (**d**) infection rate, (**e**) cure rate, and (**f**) death rate; (**g**) number of daily confirmed, (**h**) number of daily recovered, and (**i**) number of daily deaths.

**Figure 4 ijerph-20-00476-f004:**
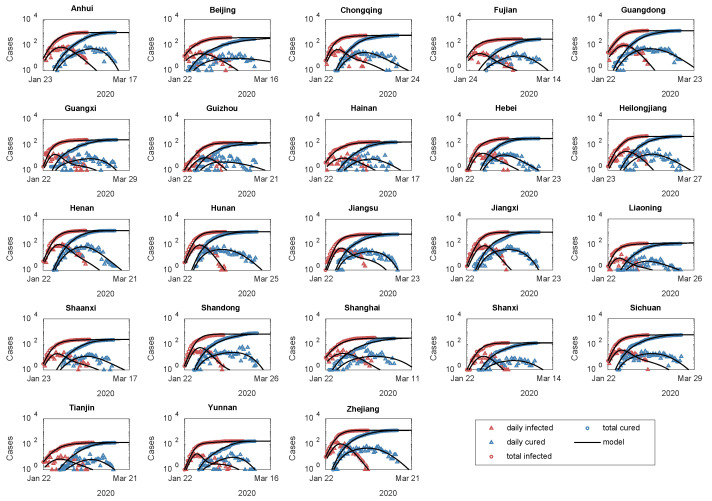
Simulation for 23 areas of mainland China in first wave of the epidemic, compared to data. Circles indicate cumulative data (as of 1 April 2020) collected from Johns Hopkins University real-time epidemic surveillance system, triangles indicate daily data calculated from that, and solid lines indicate simulation results from *SHR* model.

**Figure 5 ijerph-20-00476-f005:**
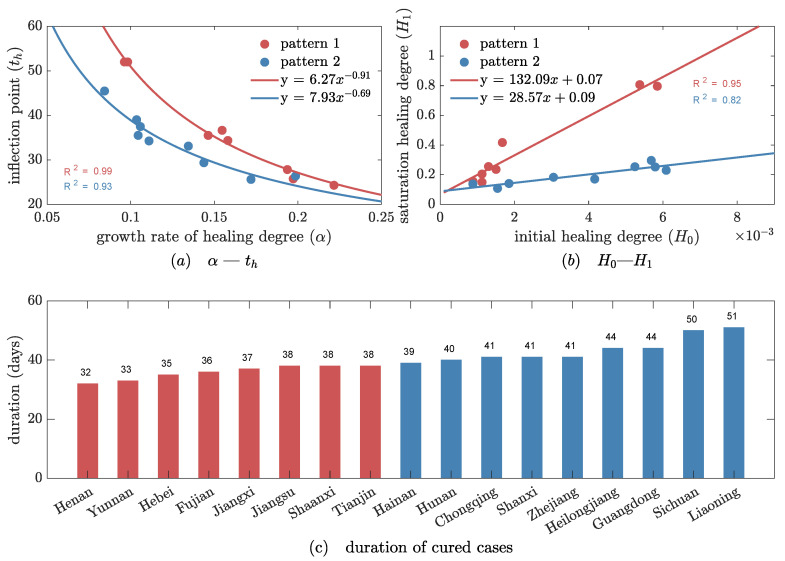
Two evolutionary patterns of healing degree among 17 areas of mainland China. Red represents pattern 1 and blue represents pattern 2. (**a**) Power laws of growth rate α and inflection point $t_{h}$. (**b**) Linear laws of initial healing degree $H_{0}$ and saturation healing degree $H_{1}$. (**c**) Distribution of duration.

**Figure 6 ijerph-20-00476-f006:**
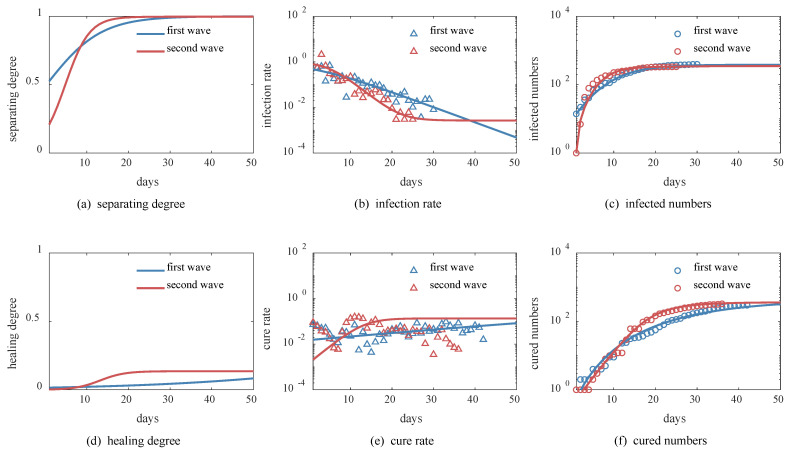
Comparison of Beijing in different epidemic waves. Red and blue circles indicate data collected by Johns Hopkins University real-time epidemic surveillance system (as of 2 July 2020). Triangles indicate resulting calculated rate data, and solid lines indicate results of *SHR* model simulations. (**a**) Separating degree, (**b**) infection rate, and (**c**) number of infected; (**d**) healing degree, (**e**) cure rate, and (**f**) number of cured.

**Figure 7 ijerph-20-00476-f007:**
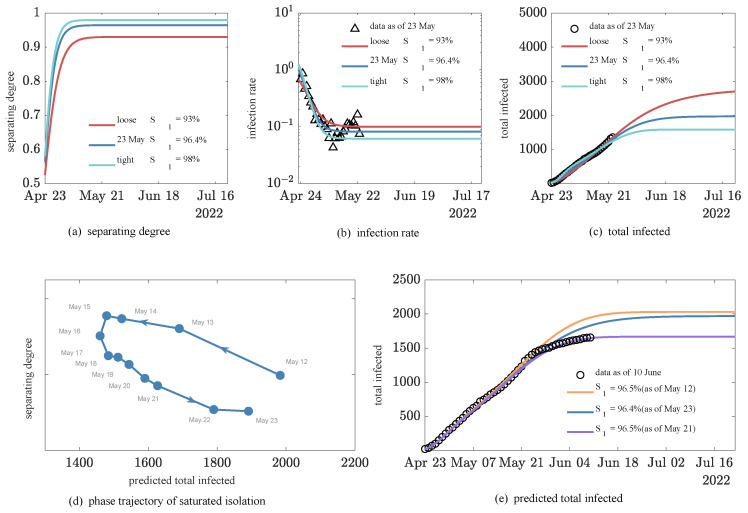
Third epidemic wave in Beijing under different intervention intensities. Black circles indicate data collected from Johns Hopkins University real-time epidemic surveillance system, triangles indicate resulting rate data, and solid lines indicate *SHR* model simulations, with different colors representing different saturation control efforts. (**a**) Evolution of separating degree, (**b**) infection rate, and (**c**) total predicted infected cases; (**d**) phase trajectory of separating degree by simulating with different sized datasets; (**e**) predicted evolution of infected cases by using different sized datasets.

**Table 1 ijerph-20-00476-t001:** Model parameters and initial values for Hubei (China) and Italy.

Order Parameters	Model Parameters		Hubei	Italy
Separating degree (S)	$V_{I}^{0}$	Initial infection rate	0.48 ± 0.01	0.48 ± 0.02
	γ	Growth rate of separating degree	0.18 ± 0.02	0.10 ± 0.02
	$S_{1}$	Saturation value of separating degree	1.00 ± 0.00	0.98 ± 0.01
	$t_{s}$	Time up to midpoint of S	17.8 ± 0.4	17.0 ± 1.0
Healing degree (H)	α	Growth rate of healing degree	0.08 ± 0.01	0.03 ± 0.01
	$H_{1}$	Saturation value of healing degree	0.15 ± 0.01	0.09 ± 0.04
	$t_{h}$	Time up to midpoint of H	50 ± 2	96 ± 20
Rescuing degree (R)	$V_{D}^{0}$	Initial death rate	0.03 ± 0.01	0.03 ± 0.01
	μ	Growth rate of rescuing degree	0.16 ± 0.06	0.09 ± 0.03
	$R_{1}$	Saturation value of rescuing degree	0.95 ± 0.02	0.80 ± 0.10
	$t_{r}$	Time up to midpoint of R	9 ± 5	32 ± 18

**Table 2 ijerph-20-00476-t002:** Model parameters for 23 epidemic spillover areas during first wave in mainland China.

γ	$t_{s}$	α	$t_{h}$	Infected	Duration
0.27 ± 0.09	6.5 ± 2.7	0.12 ± 0.05	43.4 ± 18.9	528 ± 395	27 ± 5

**Table 3 ijerph-20-00476-t003:** Comparison of model parameters and statistics for the two epidemic waves.

Wave	γ	$S_{0}$	α	Infected	Duration
First	0.15	53%	0.04	395	30
Second	0.35	21%	0.34	335	25

**Table 4 ijerph-20-00476-t004:** Mean values of model parameters obtained based on different sized datasets.

γ	$S_{0}$	$S_{1}$	α	$H_{0}$	Infected
0.33 ± 0.01	56.2% ± 0.1%	96.5% ± 0.1%	0.071 ± 0.005	1.8% ± 0.1%	1631 ± 166

## Data Availability

All data and code used in this study are publicly available at https://github.com/zhanglei-pku/COVID-19-SHR-model (accessed on 26 December 2022).

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
