# Peer review of "Quantifying Social Interventions for Combating COVID-19 via a Symmetry-Based Model"

_ijerph, 2022, doi:10.3390/ijerph20010476_

Round 1

Reviewer 1 Report

This paper is a valuable addition to epidemiology. In the past two years a large amount of data has been collected and it is high time researchers revisiting many of the fundamental assumptions in the mathematical models of epidemics. This paper can be seen as a first step in this direction, through inventing phenomenological models that fits the data. 

The paper is very well-organized and well-written. I suggest publication in the present form. 

Author Response

Thanks for the encouraging comments.

Reviewer 2 Report

The manuscript is organized well and contains solid contributions, can be accepted for publication if the authors respond to the following comments properly.

Author Response

Thanks for the encouraging comments. Please see the attachment for details. 

Author Response

We are very grateful to the reviewer for your careful investigation and constructive suggestions for this article, which is very helpful for us to express relevant content more accurately. Please see the attachment for details.

Round 2

Author Response

We are very grateful to the reviewer for your careful investigation and constructive suggestions for this article, which is very helpful for us to express relevant content more accurately.
